# A secretory protein neudesin regulates splenic red pulp macrophages in erythrophagocytosis and iron recycling

Yoshiaki Nakayama [1], Yuki Masuda[1], Takehiro Mukae[1], Tadahisa Mikami [2], Ryohei Shimizu[1], Naoto Kondo[1], Hiroshi Kitagawa [2], Nobuyuki Itoh[3] & Morichika Konishi [1]✉

Neudesin, originally identified as a neurotrophic factor, has primarily been studied for its neural functions despite its widespread expression. Using 8-week-old *neudesin* knockout mice, we elucidated the role of neudesin in the spleen. The absence of *neudesin* caused mild splenomegaly, shortened lifespan of circulating erythrocytes, and abnormal recovery from phenylhydrazine-induced acute anemia. Blood cross-transfusion and splenectomy experiments revealed that the shortened lifespan of erythrocytes was attributable to splenic impairment. Further analysis revealed increased erythrophagocytosis and decreased iron stores in the splenic red pulp, which was linked to the upregulation of Fcγ receptors and iron-recycling genes in *neudesin*-deficient macrophages. In vitro analysis confirmed that neudesin suppressed erythrophagocytosis and expression of Fcγ receptors through ERK1/2 activation in heme-stimulated macrophages. Finally, we observed that 24-week-old *neudesin* knockout mice exhibited severe symptoms of anemia. Collectively, our results suggest that neudesin regulates the function of red pulp macrophages and contributes to erythrocyte and iron homeostasis.

[1] Laboratory of Microbial Chemistry, Kobe Pharmaceutical University, Kobe, Japan. [2] Laboratory of Biochemistry, Kobe Pharmaceutical University, Kobe, Japan. [3] Kyoto University Graduate School of Pharmaceutical Science, Kyoto, Japan. ✉email: mkonishi@kobepharma-u.ac.jp

After development in the bone marrow, red blood cells (RBCs) enter the circulatory system and transport oxygen throughout the body. The structural flexibility and deformability of the RBC membrane are critical for transport through narrow blood capillaries, but they gradually decrease with age owing to physical and oxidative stresses[1,2]. Therefore, the body employs a mechanism to remove aged RBCs from circulation. After circulating for approximately 120 days in humans and 30 days in mice, aged RBCs are captured mainly by red pulp macrophages (RPMs) in the red pulp of the spleen. These macrophages digest aged RBCs and release heme-associated iron, which is recycled into new hemoglobin during erythropoiesis in the bone marrow[3].

Several biological factors are implicated in the removal of aged RBCs. So-called "eat me" signals emerge on the surface of aged RBCs, triggering their clearance[1]. First, oxidative stress induces clustering of membrane protein band3, an abundant anion-transporter protein of RBCs[1]. The cluster is detected by naturally occurring anti-band3 autoantibodies, and complement activation through the classical pathway further enhances the opsonization of aged RBCs. This complex is recognized by Fcγ receptors (FcγRs) and complement receptor CR3 (Mac-1), which comprises CD11b and CD18 on the cell membrane of macrophages and induces erythrophagocytosis. As the second "eat me" signal, phosphatidylserine (PS) is exposed on the outer surface of RBCs[2]. Similar to other cell types, PS is located on the inner leaflet of healthy RBCs; however, under oxidative stress, osmotic shock, or other forms of cell stress, PS is exposed and accumulated in the outer leaflet of RBCs. Macrophages recognize PS-presenting RBCs with PS receptors, including Stabilin-2, Tim-1, and Tim-4[4,5]. TAM receptors (Tyro3, Axl, Mer) also recognize PS through mediation by the bridging molecule Gas-6[1]. The recognition of PS exposure by these receptors leads to phagocytosis of aged RBCs. Meanwhile, a "don't eat me" signal functions as a suppressor of erythrophagocytosis. CD47 expressed on the surface of healthy RBCs binds to signal-regulatory protein alpha (SIRPα) present on macrophages, inhibiting erythrophagocytosis[6]. However, oxidative stress leads to a conformational change in the CD47 molecule, thereby switching on the "eat me" signal and inducing phagocytic signals through SIRPα, resulting in the clearance of aged RBCs[7].

We previously identified and isolated a heme-binding secreted protein, neudesin, which is expressed in several tissues with the highest abundance in the central nervous system, with in vitro neurotrophic activity via the activation of intracellular signaling pathways[8,9]. Neudesin contains a heme/steroid-binding domain highly similar to cytochrome b5, which maintains the reduced state of hemoglobin in erythrocytes, and its neurotrophic activity is dependent on heme binding to this domain[10]. Although its receptor has not yet been identified, it is expected to be a G protein-coupled receptor (GPCR) owing to the inhibition of neudesin activity by inhibitors of the G protein-coupled signaling pathway[8,9,11]. In the adult mouse brain, neudesin is involved in appetite, anxiety behavior, and contextual fear memory[12–14]. Furthermore, neudesin has been increasingly studied in non-neural tissues. To date, our group demonstrated the functions of neudesin in adipose tissues[15,16], and another group reported that neudesin regulates bovine adipogenesis and myogenesis[17]. Pathophysiologically, neudesin is also involved in tumorigenesis[18–20].

In addition to its expression in neural and adipose tissues, neudesin mRNA is expressed in the spleen, although its splenic effects remain to be elucidated. In this study, we demonstrated the function of neudesin in the spleen. Eight-week-old neudesin knockout (KO) mice displayed abnormal RBC accumulation in the spleen. Circulating erythrocytes from neudesin KO mice showed enhanced resistance against hemolytic conditions in vitro as well as a shortened lifespan in vivo. These erythrocyte phenotypes were largely attributable to the hyperactivation of erythrophagocytosis by RPMs lacking neudesin, presumably owing to upregulation of FcγRs on RPMs of neudesin KO mice. Neudesin deficiency also led to impaired iron homeostasis in the spleen and incomplete recovery from drug-induced acute anemia. After maturation, 24-week-old neudesin KO mice exhibited normocytic anemia associated. Collectively, neudesin contributes to the homeostasis of erythrocytes and iron by regulating macrophage functions.

## Results

### Neudesin-deficient mice have more erythrocytes in the spleen and abnormalities in circulating erythrocytes

In addition to the brain and white adipose tissue where neudesin's function has been previously reported, neudesin is also expressed in the spleen (Fig. 1a). However, its function in the spleen is not clear. Therefore, we first examined the spleens of 8-week-old WT and neudesin KO mice. The spleens of neudesin KO mice were modestly larger and slightly (but significantly) heavier than those of WT mice (Fig. 1b, c), suggesting that neudesin influenced spleen contents. Therefore, we investigated which type of cells contributed to mild splenomegaly in neudesin KO mice. Flow cytometric analysis showed that the spleens of neudesin KO mice contained approximately 1.5-fold more TER119+ RBCs compared with those of WT mice (Fig. 1d). The number of CD45+ white blood cells in the spleens was comparable between WT and neudesin KO mice. These data suggest that neudesin is involved in RBC homeostasis, as in the case of cytochrome b5 reductase, which has a heme-binding domain similar to the heme/steroid-binding domain of neudesin[10]. To assess this possibility, peripheral blood was hematologically assessed, revealing almost normal RBCs except for a slight reduction in mean corpuscular hemoglobin (MCH) and an increase in platelet count in neudesin KO mice (Table 1). Additionally, the RBCs of neudesin KO mice were significantly more resistant to hypertonic and sugar-depleted hemolytic conditions (Fig. 1e). These data indicated that the RBCs of neudesin KO mice were more deformable. Because the deformability of erythrocytes is used as an indicator of cellular senescence[1], circulating erythrocytes in neudesin KO mice may be a younger population than those in WT mice. We further measured the ratios of PS exposure in erythrocytes and naturally occurring antibody-bound erythrocytes, which revealed no significant difference between WT and neudesin KO mice (Fig. 1f, g). These results showed that there was little change in the percentage of aged erythrocytes between WT and neudesin KO mice.

### Erythrocytes in neudesin KO mice have a shortened lifespan owing to impairment of the spleen rather than of erythrocytes

Our data suggest that neudesin is implicated in the senescence of RBCs. Accordingly, we examined the lifespan of RBCs in WT and neudesin KO mice using direct in vivo biotinylation of RBCs. Sulfo-NHS-biotin was injected intravenously to label RBCs, and the lifespan of RBCs was determined by flow cytometric analysis of peripheral biotinylated RBCs (Fig. 2a, b). The half-life of labeled erythrocytes in WT mice was approximately $17.2 \pm 3.0$ days; in contrast, that in neudesin KO mice decreased to about $13.3 \pm 1.7$ days (Fig. 2c).

Because neudesin KO mice had a shortened lifespan of erythrocytes under normal conditions, we compared erythrocytes from WT and neudesin KO mice during acute anemia induced by phenylhydrazine injection. Phenylhydrazine injection decreased RBC counts and hematocrit levels, which recovered 4 and 8 days after injection in WT neudesin KO and mice, respectively (Fig. 2d, f). Neudesin KO mice showed impaired

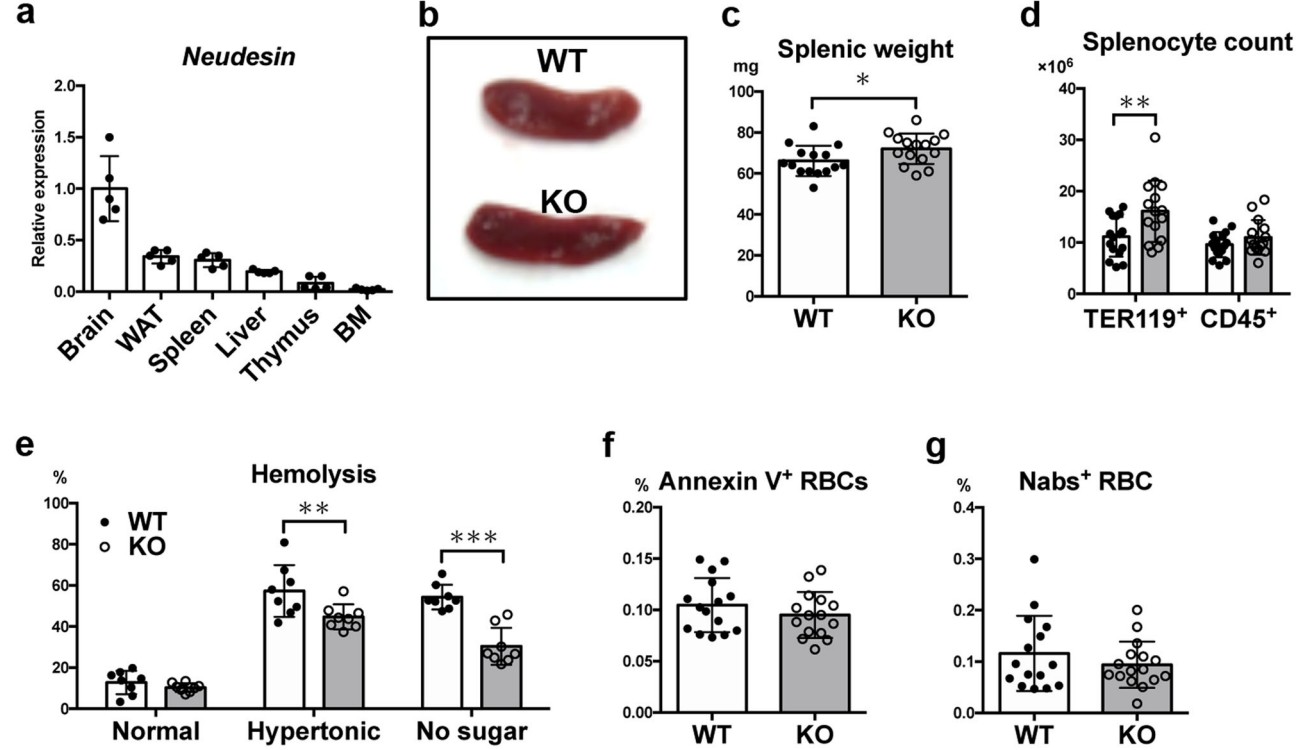

**Fig. 1 *Neudesin*-deficient mice exhibit mild splenomegaly and abnormalities in circulating erythrocytes. a** Relative *neudesin* mRNA levels in tissues of 8-week-old C57BL/6 male mice ($n = 5$) were quantified by RT-qPCR. WAT, white adipose tissue. BM, bone marrow. **b** A representative picture of spleens from 8-week-old wildtype (WT) and *neudesin* knockout (KO) male mice. **c** Spleens of *neudesin* KO mice ($n = 15$, male) were significantly heavier than those of 8-week-old WT male mice ($n = 15$, $P = 0.0393$ by unpaired $t$ test). **d** Splenocyte suspensions from 8-week-old WT ($n = 16$) and *neudesin* KO male mice ($n = 15$) were assessed by flow cytometric analysis using anti-TER119 and anti-CD45 mAbs, and cell numbers of TER119$^+$ red blood cells and CD45$^+$ white blood cells in the spleen were counted. $P = 0.0098$, and $0.1862$, respectively, by unpaired $t$ test. **e** Hemolysis ratios were measured after incubating blood cells from WT and KO male mice for 16 h in Ringer's solution (left), hypertonic solution (middle), or sugar-depleted solution (right). $P = 0.8905$, $0.0059$, and $<0.0001$, respectively, by unpaired t-test ($n = 8$ for each group). Exposure of phosphatidylserine on the outer plasma membrane of erythrocytes was measured by annexin V binding (**f**) ($n = 15$ for each group, male, $P = 0.2840$), and naturally occurring antibody-bound erythrocytes were detected with anti-mouse IgG and IgM antibodies (**g**) (WT: $n = 15$, KO: $n = 16$, male, $P = 0.3183$).

**Table 1 Young *neudesin*-deficient mice have slightly lower MCH and higher PLT levels than wild-type mice.**

| Test | WT | KO | P |
|---|---|---|---|
| RBC (×10⁴/μL) | 870 ± 12, $n = 17$ | 904 ± 13, $n = 18$ | 0.079 |
| Hb (g/dL) | 14.0 ± 0.2, $n = 17$ | 14.2 ± 0.2, $n = 18$ | 0.448 |
| Ht (%) | 47.2 ± 0.8, $n = 17$ | 48.3 ± 0.8, $n = 18$ | 0.335 |
| MCH (pg) | 16.1 ± 0.13, $n = 17$ | 15.7 ± 0.06, $n = 18$ | 0.013* |
| MCV (fL) | 53.8 ± 0.20, $n = 17$ | 53.4 ± 0.28, $n = 18$ | 0.281 |
| WBC (/μL) | 3088 ± 480, $n = 17$ | 3522 ± 450, $n = 18$ | 0.882 |
| PLT (×10⁴/μL) | 86.8 ± 8.6, $n = 17$ | 107.2 ± 4.4, $n = 18$ | 0.039* |
| Reticulocytes (%) | 1.71 ± 0.06, $n = 22$ | 1.62 ± 0.07, $n = 19$ | 0.301 |

Data are shown as means ± SD of three or more experiments from WT and Neudesin KO mice at 8 weeks of age.
*RBC* red blood cell count, *Hb* hemoglobin, *Ht* hematocrit, *MCH* mean corpuscular hemoglobin, *MCV* mean corpuscular volume, *WBC* white blood cell count, *PLT* platelet count.
*$P < 0.05$ by unpaired t-test.

hemoglobin concentrations after injection (Fig. 2e). *Neudesin* mRNA levels were barely detected in the bone marrow but remarkably increased in the spleens of WT mice after induction of acute anemia (Fig. 2g). These data suggest that neudesin is necessary for iron recycling in acute anemic conditions.

To further investigate the cause of the shortened lifespan of RBCs in *neudesin* KO mice, we performed a blood cross-transfusion experiment. Biotinylated WT or *neudesin* KO RBCs were transfused into WT or *neudesin* KO mice (Fig. 2h). Consequently, mice in both *neudesin* KO groups demonstrated accelerated clearance of biotinylated RBCs compared with that of

WT mice transfused with the same RBCs (Fig. 2i). These results suggest that some environments surrounding erythrocytes rather than the properties of the erythrocytes themselves cause the shortened lifespan of RBCs in *neudesin* KO mice.

The splenomegaly phenotype in *neudesin* KO mice implies that impaired spleen function leads to a shortened erythrocyte lifespan in *neudesin* KO mice. Thus, we examined the effect of splenectomy on erythrocyte lifespan (Fig. 2j). In splenectomized *neudesin* KO mice, the lifespan of erythrocytes was recovered equally to that of splenectomized WT mice, while the erythrocyte lifespan of sham KO mice remained shorter than that of sham

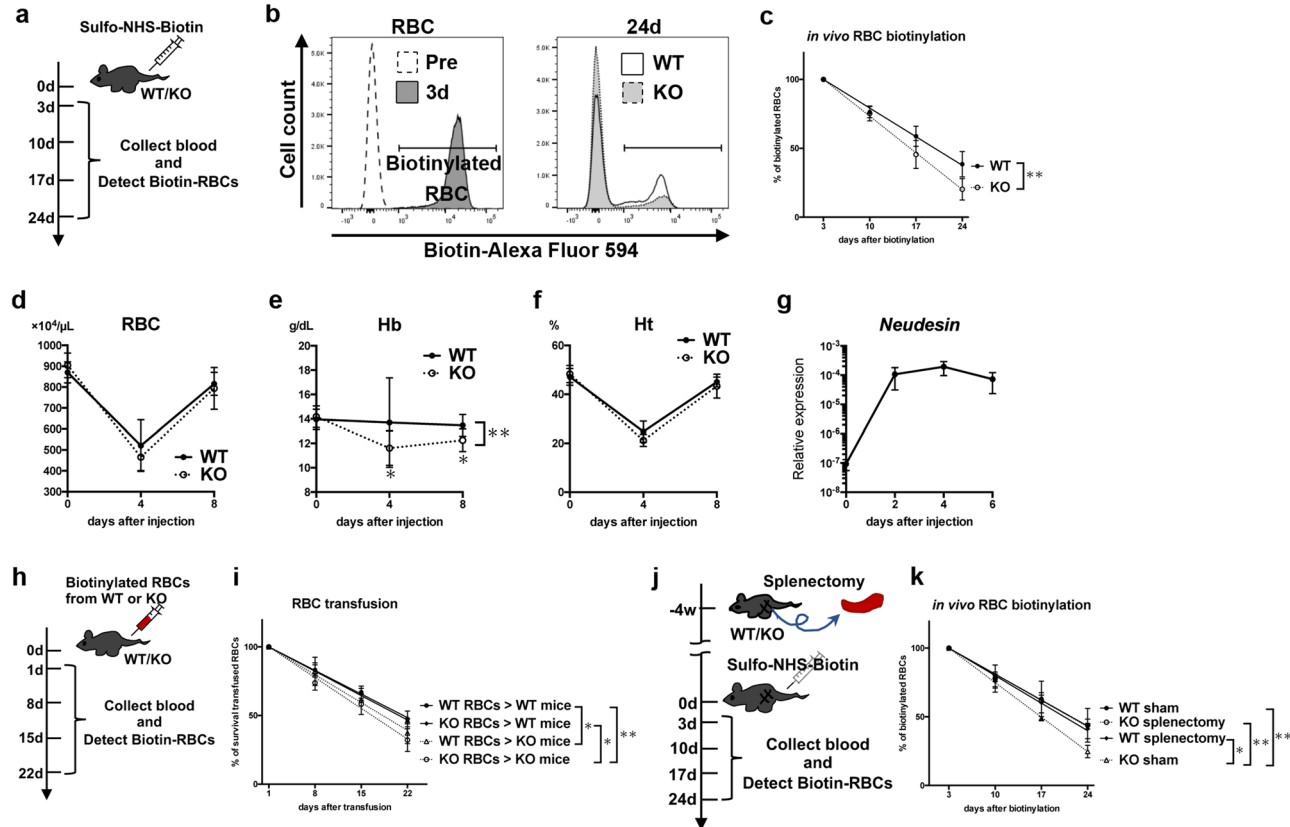

**Fig. 2 Erythrocytes in *neudesin* KO mice have a shortened lifespan owing to the spleen rather than erythrocytes themselves. a** Scheme of experiments to measure erythrocyte lifespan in vivo using intravenous administration of sulfo-NHS-biotin at 8 weeks of age. RBCs were collected from tail veins on days 3, 10, 17, and 24 after administration. Biotinylated RBCs were detected using fluorescently conjugated streptavidin. **b** More than 99% of RBCs were biotinylated on day 3 after administration (3 d; gray histogram in the left panel). The right panel shows representative histograms of WT (white) and *neudesin* KO (gray) on day 24 after administration (24 d). **c** The percentage of biotinylated RBCs from WT and *neudesin* KO male mice was tracked by flow cytometry. WT: $n = 12$, KO: $n = 10$. $P = 0.0010$. Blood samples were collected from WT ($n \geq 7$) and *neudesin* KO ($n \geq 9$) male mice on days 0, 4, and 8 after intraperitoneal administration of phenylhydrazine at 8 weeks of age. RBC counts (**d**) ($P = 0.4199$), hemoglobin concentrations (Hb) (**e**) ($P = 0.0019$), and hematocrit levels (Ht) (**f**) ($P = 0.1057$) were measured. **g** Relative *neudesin* mRNA levels in spleens of WT male mice ($n = 3$) were quantified by RT-qPCR on days 0, 2, 4, and 6 after intraperitoneal administration of phenylhydrazine. **h** The scheme of experiments to measure the lifespan of RBCs transfused from WT or *neudesin* KO male mice into WT or *neudesin* KO male mice at 8 weeks of age. RBCs were collected on days 1, 8, 15, and 22 after transfusion. **i** The percentage of survival of transfused RBCs was tracked by flow cytometry. WT RBCs transfused into WT mice: $n = 10$, WT RBCs into KO mice: $n = 6$, KO RBCs into WT mice: $n = 5$, KO RBCs into KO mice: $n = 6$. (WT RBCs > KO mice vs. WT RBCs > WT mice: $P = 0.0260$, KO RBCs > WT mice vs. WT RBCs > WT mice: $P > 0.9999$, KO RBCs > KO mice vs. WT RBCs > WT mice: $P < 0.0001$, KO RBCs > WT mice vs. WT RBCs > KO mice: $P = 0.3728$, KO RBCs > KO mice vs. WT RBCs > KO mice: $P = 0.9843$, KO RBCs > KO mice vs. KO RBCs > WT mice: $P = 0.0077$). **j** The scheme of experiments to measure the lifespan of erythrocytes 4 weeks after splenectomy or sham-splenectomy. RBCs were collected on days 3, 10, 17, and 24 after administration. **k** The percentage of biotinylated RBCs from splenectomized and sham-splenectomized male mice was tracked by flow cytometry. WT sham: $n = 4$, KO sham: $n = 4$, WT splenectomy: $n = 6$, KO splenectomy: $n = 6$. (KO sham vs. WT sham: $P = 0.0087$, WT splenectomy vs. WT sham: $P = > 0.9999$, KO splenectomy vs. WT sham: $P = > 0.9999$, WT splenectomy vs. KO sham: $P = 0.0117$, KO splenectomy vs. KO sham: $P = 0.0016$, KO splenectomy vs. WT splenectomy: $P = > 0.9999$) Data are shown as means ± SD of two or more experiments. *$P < 0.05$ and **$P < 0.01$ by a 2-way analysis of variance followed by Sidak's post-test.

WT mice (Fig. 2k). These data indicate that the shortened lifespan of RBCs in *neudesin* KO mice might be caused by increased removal of aged RBCs in the spleen.

**Enhanced erythrophagocytosis and decreased iron accumulation by RPM lacking *neudesin*.** The spleen is known to recognize and remove old, malformed, and damaged erythrocytes. RPMs are the primary phagocytes that engulf these erythrocytes in the spleen. To investigate whether neudesin was involved in the phagocytic function of RPM, we first examined RPM populations in *neudesin* KO mice. The ratios, absolute cell numbers, and distributions of RPMs showed little difference between WT and *neudesin* KO mice (Fig. 3a–c). Next, to investigate the

erythrophagocytic activity of RPMs, erythrocytes were labeled ex vivo with PKH26 fluorescent dye and traced in WT and *neudesin* KO mice (Fig. 3d–f). Three days after PKH26+ RBCs were administered, about 50% of WT RPMs had engulfed PKH26+ RBCs (Fig. 3e, f). In *neudesin* KO mice, the ratio of PKH+ RPMs increased significantly to about 60% (Fig. 3f). Considering that *neudesin* mRNA was relatively abundant in F4/80+ RPMs (Fig. 3g), it can be assumed that neudesin suppresses phagocytic activity in an autocrine/paracrine manner. In addition to erythrophagocytosis, RPMs were involved in the accumulation and distribution of iron. We examined the involvement of neudesin in iron homeostasis. Perl's Prussian staining, which specifically stains iron, revealed decreased iron accumulation in *neudesin* KO RPMs (Fig. 3h). Quantitative determinations of the

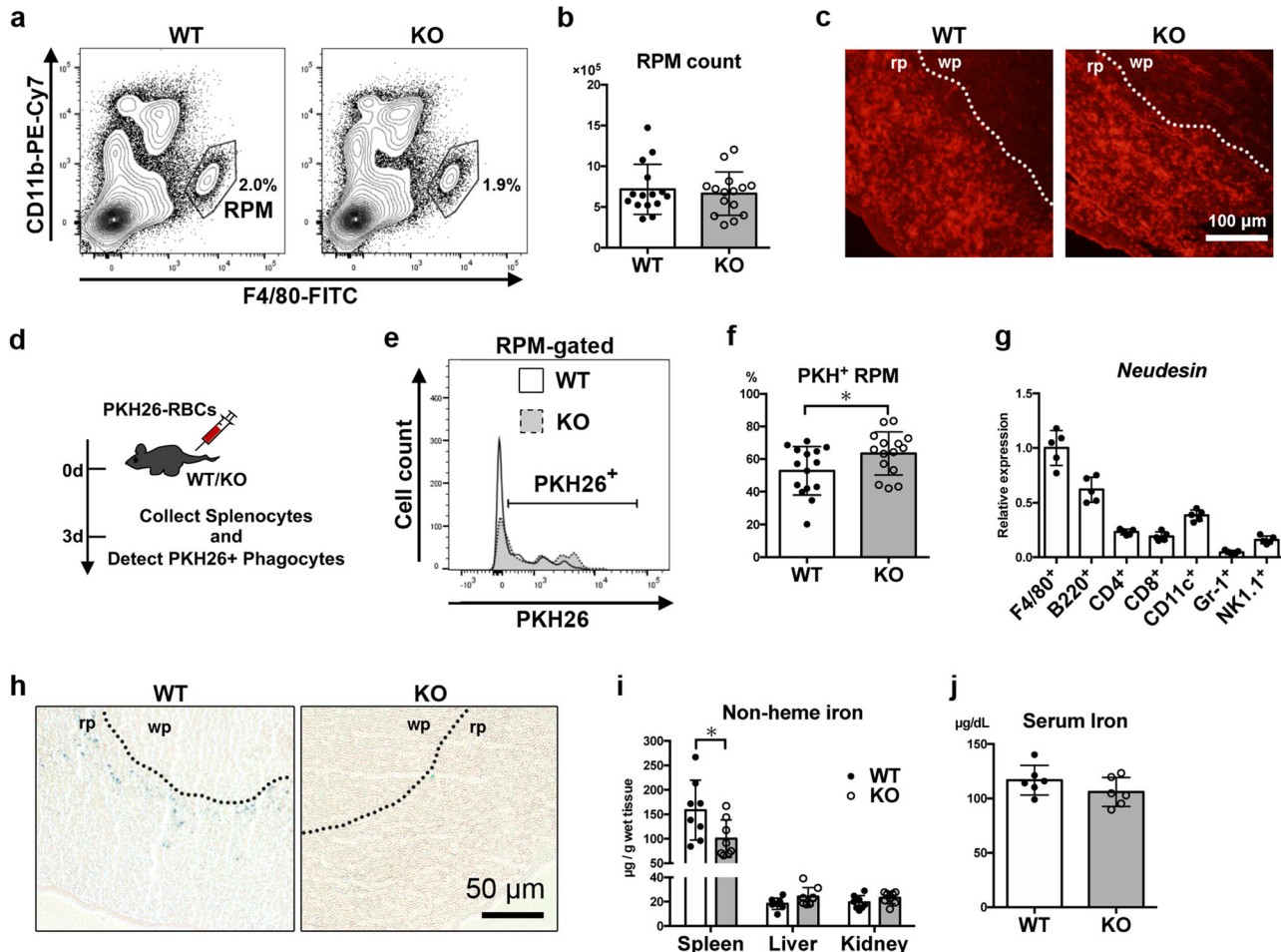

**Fig. 3 Enhanced erythrophagocytosis by red pulp macrophages and decreased iron accumulation in spleen lacking *neudesin*. a** Splenocytes from 8-week-old WT and *neudesin* KO male mice were stained with anti-F4/80 and anti-CD11b mAbs. Representative flow cytometry plots show populations of red pulp macrophages (RPM). **b** Chart showing the number of RPMs from 8-week-old WT and *neudesin* KO male mice (WT: $n = 15$, KO: $n = 15$, $P = 0.6308$). **c** Autofluorescence of spleen sections from 8-week-old WT and *neudesin* KO male mice were observed to visualize RPMs. Dotted lines indicate boundaries between red pulp (rp) and white pulp (wp). **d** Scheme of experiments to measure erythrophagocytosis in vivo using transfusion of PKH26-conjugated RBCs into 8-week-old WT and KO male mice. On day 3 after transfusion, splenocytes were collected and analyzed by flow cytometry to detect PKH26+ RPMs. **e** Representative histograms of RPMs from 8-week-old WT (white) and KO (gray) male mice. **f** The percentage of PKH+ RPMs from 8-week-old WT and *neudesin* KO male mice tracked by flow cytometry. *$P < 0.05$ by unpaired t-test (WT: $n = 15$, KO: $n = 15$, $P = 0.0469$). **g** Splenocytes from 8-weeks old male mice were sorted with lineage-specific antibodies, and relative *neudesin* mRNA expression levels were quantified by RT-qPCR ($n = 5$). **h** Spleen sections from 8-week-old WT and *neudesin* KO male mice were stained with Perl's Prussian blue to visualize the iron deposits. Dotted lines indicate boundaries between red pulp (rp) and white pulp (wp). **i** Non-heme iron content was quantified in spleens, livers, and kidneys from 8-week-old WT and *neudesin* KO male mice. Each symbol represents an individual mouse. Spleen: $P = 0.0383$, liver: $P = 0.0737$, and kidney: $P = 0.1793$ by unpaired t-test ($n = 8$). **j** The graph shows serum iron levels of 8-week-old WT and *neudesin* KO male mice. $P = 0.1924$ by unpaired t-test ($n = 6$) Each symbol represents an individual mouse. Data are shown as means ± SD of three or more experiments.

non-heme iron contents in spleens, livers, kidneys, and blood serum revealed that splenic iron content reduced by ~40% in *neudesin* KO mice (Fig. 3i, j).

**FcγR upregulation on the surface of RPM in *neudesin* KO mice.** Erythrophagocytosis is triggered by several types of phagocytic receptors on the surface of RPMs[1]. We, therefore searched for receptors that mediate accelerated erythrophagocytosis in *neudesin* KO mice. The quantification of receptors by flow cytometry revealed the upregulation of FcγRs (FcγR1, FcγR2/3, and FcγR4) on the surface of RPMs in *neudesin* KO mice, whereas the expression of receptors recognizing PS (Stabilin-2, Tim-4, and Axl) remained unchanged (Fig. 4a, b). Furthermore, the levels of CD163 molecules, which are erythroblast adhesion receptors[21], also increased in the *neudesin*

KO RPMs. To investigate whether the upregulation of FcγRs and CD163 is induced at the transcriptional level in *neudesin* KO RPMs, RPMs from WT and *neudesin* KO mice were isolated with anti-F4/80 microbeads, and *Adgre1*-coding F4/80 molecules, *Fcgr1, Fcgr2b, Fcgr3, Fcgr4*, and *CD163* mRNAs were quantified (Fig. 4c). In *neudesin* KO RPMs, *Fcgr4* and *CD163* mRNAs were significantly enhanced compared with those in WT RPMs. The expression levels of *Fcgr1, Fcgr2b, and Fcgr3* genes tended to increase in *neudesin* KO RPMs without significance. *Hmox1* and *Slc40a1*, which code for heme oxygenase 1 and iron transporter Ferroportin-1, respectively, were also upregulated in *neudesin* KO RPMs, suggesting the possibility of enhanced heme catabolism and export after digestion of RBCs (Fig. 4d). *Spic*, which controls the development of red pulp macrophages[22], was also increased in *neudesin* KO RPMs (Fig. 4d).

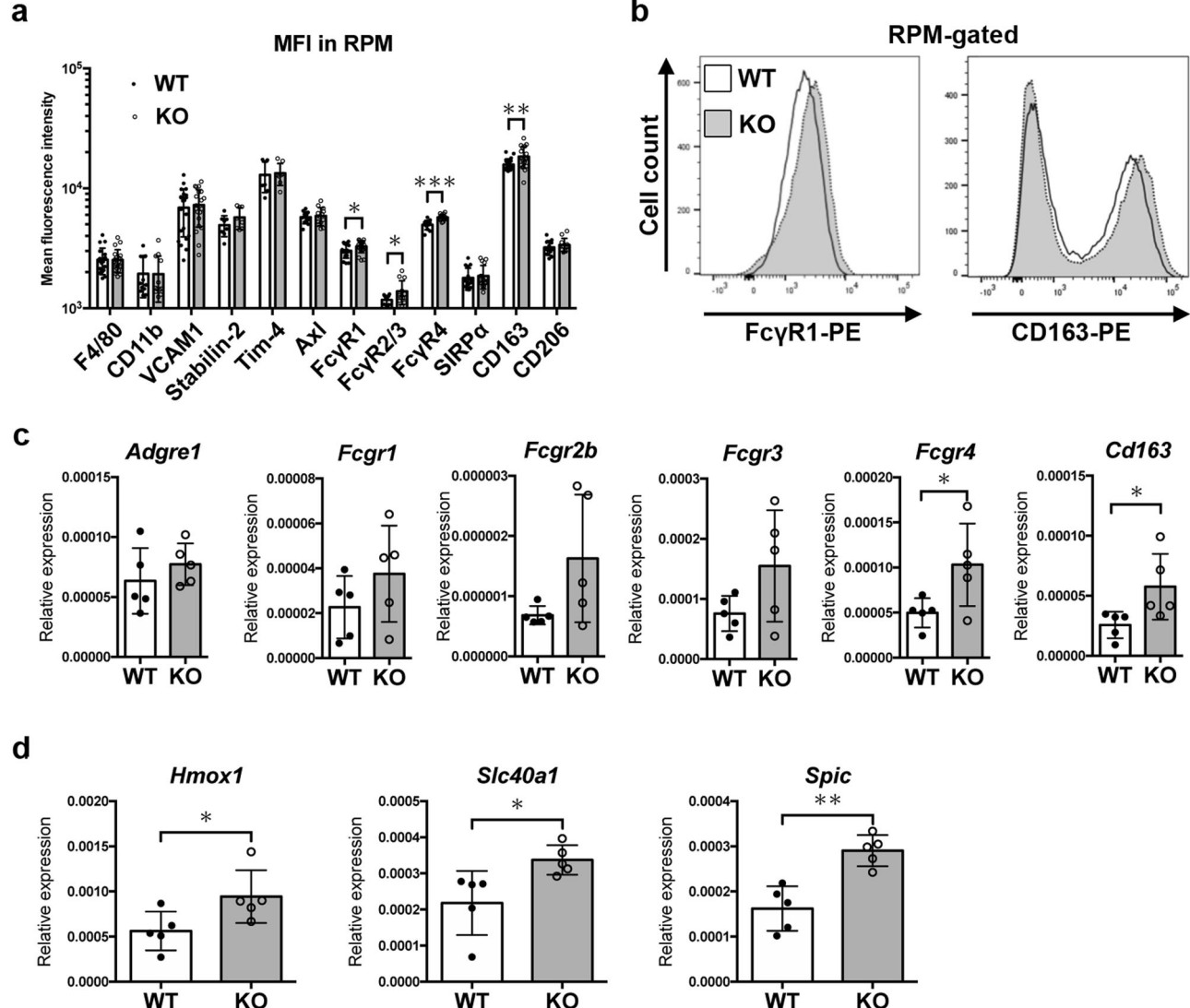

**Fig. 4 FcγRs, CD163, and iron-recycling genes are upregulated in red pulp macrophages in *neudesin* KO mice. a** The mean fluorescence intensity (MFI) of surface receptors on the gated red pulp macrophages (RPM) from 8-week-old WT and *neudesin* KO male mice was measured by flow cytometry ($n \geq 6$, F4/80: $P = 0.9388$, CD11b: $P = 0.9629$, VCAM1: $P = 0.6770$, Stabilin-2: $P = 0.1617$, Tim-4: $P = 0.7969$, Axl: $P = 0.6595$, FcγR1: $P = 0.0370$, FcγR2/3: $P = 0.0356$, FcγR4: $P < 0.0001$, SIRPα: $P = 0.6267$, CD163: $P = 0.0064$, and CD206: $P = 0.3749$ by unpaired $t$ test) **b** Representative histograms of RPMs treated with anti-FcγR1 (left panel) and anti-CD163 (right panel) from 8-week-old WT (white) and *neudesin* KO (gray) male mice. **c** RPMs from 8-week-old WT and KO male mice were isolated with magnetic bead-conjugated anti-F4/80 antibody, and relative mRNA expression levels of genes encoding macrophage receptors were quantified by RT-qPCR ($n = 5$ each groups, *Adgre1*: $P = 0.3676$, *Fcgr1*: $P = 0.2295$, *Fcgr2b*: $P = 0.0848$, *Fcgr3*: $P = 0.1061$, *Fcgr4*: $P = 0.0396$, and *Cd163*: $P = 0.0421$ by unpaired t-test). **d** Relative mRNA expression levels of genes regulating iron recycling were quantified by RT-qPCR with isolated RPMs from 8-week-old WT and KO male mice ($n = 5$ each groups, *Hmox1*: $P = 0.0465$, *Slc40a1*: $P = 0.0260$, and *Spic*: $P = 0.014$ by unpaired t-test). Data are shown as the mean ± SD of four or more experiments. Each symbol represents an individual mouse. *$P < 0.05$ and **$P < 0.01$, and ***$P < 0.001$ by unpaired t-test.

**Neudesin suppresses erythrophagocytosis and FcγRs expression through ERK activation in bone marrow-derived macrophages.** To examine whether the suppressive effects of neudesin on phagocytosis are restricted during erythrophagocytosis by RPMs, the phagocytic activities of bone marrow-derived macrophages (BMDMs) were determined (Fig. 5a–c). To induce RPM-like cells, we used granulocyte-macrophage colony-stimulating factor (GM-CSF) and hemin-stimulated BMDMs because these cells have properties similar to those of RPMs[23–25]. Bone marrow cells from WT and *neudesin* KO mice were cultured in vitro in the presence of macrophage colony-stimulating (M-CSF) factor following additional stimulation with GM-CSF and hemin. The differentiation efficiency of *neudesin* KO bone marrow cells into

F4/80+ macrophages was comparable to that of WT cells (Fig. 5b). Consistent with the in vivo results, *neudesin* KO BMDMs engulfed PKH26+ RBCs more frequently than did WT BMDMs (Fig. 5c). However, WT and *neudesin* KO BMDMs phagocytosed apoptotic thymocytes and killed *E. coli* to a similar extent (Fig. 5c). Treatment with recombinant neudesin protein inhibited the enhanced erythrophagocytosis of *neudesin* KO BMDMs (Fig. 5d). These data suggest that neudesin negatively regulates phagocytosis with specificity toward erythrocytes. The slight upregulation of FcγR1 and Fcγ2/3 were also observed on the surface of *neudesin* KO BMDMs, and suppressed by the addition of recombinant neudesin protein into the culture medium (Fig. 5e). To investigate the intracellular signaling pathway

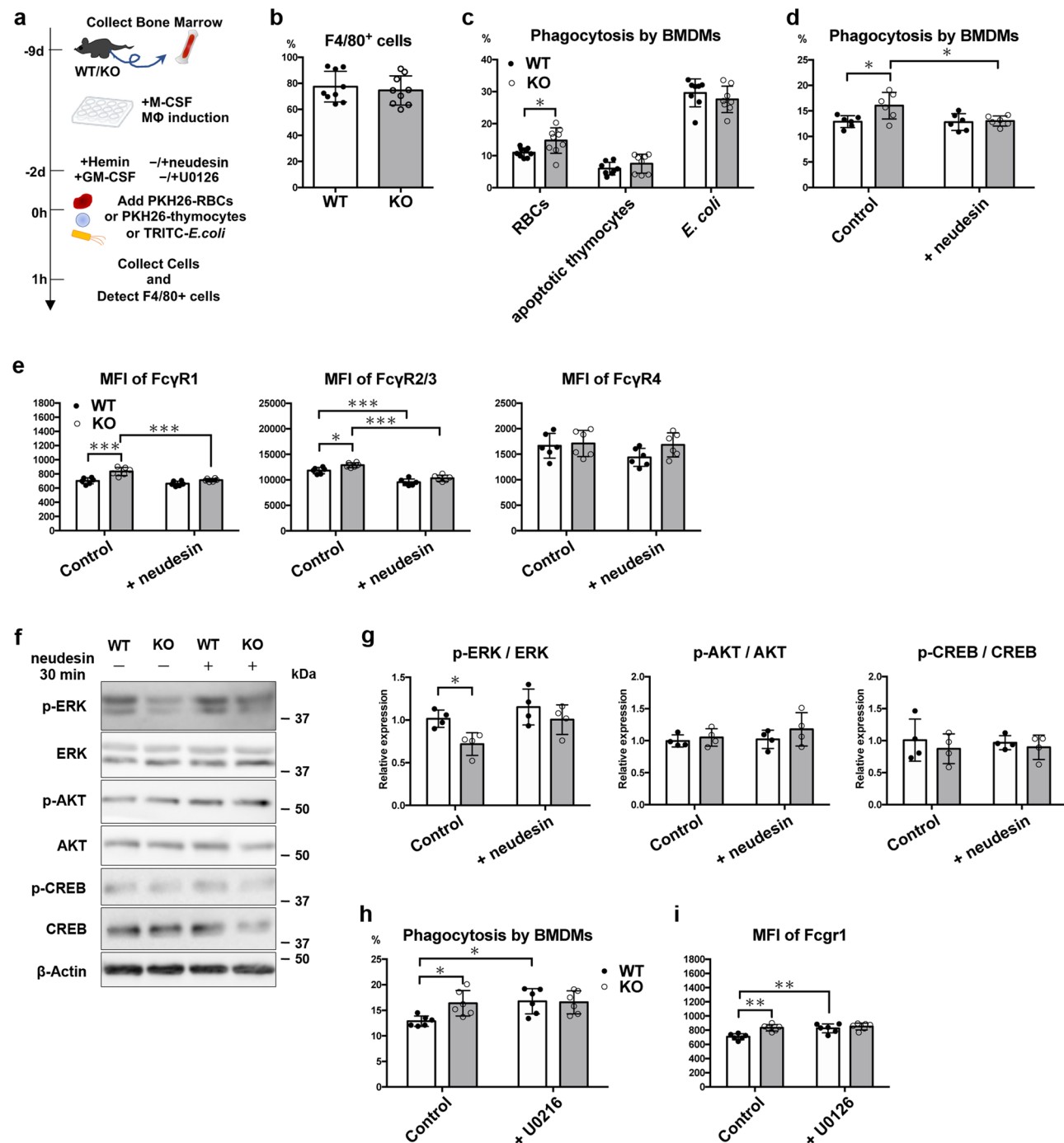

mediating the function of neudesin in BMDMs, phosphorylation of ERK1/2, AKT, and CREB, which is promoted by neudesin in neural precursor cells[9], were analyzed by western blotting. Phosphorylation of ERK1/2 was decreased in *neudesin* KO BMDMs, and upregulated by recombinant neudesin (Fig. 5f, g). AKT and CREB pathways were unchanged between WT and *neudesin* KO BMDMs and after treatment with recombinant neudesin protein. The MEK inhibitor U0126 enhanced erythrophagocytosis and surface expression of FcγR1 in WT BMDMs (Fig. 5h, i), suggesting that neudesin exerts the inhibitory effect on BMDMs through activation of ERK1/2.

**Mature *neudesin*-deficient mice exhibit normocytic anemia.** To investigate whether impaired iron recycling at a young age affects the later metabolism of erythrocytes, we examined peripheral

RBCs and splenocytes of mature WT and *neudesin* KO mice at 24 weeks of age (Fig. 6). Mature *neudesin* KO mice exhibited significantly decreased RBC counts, hemoglobin concentrations, and hematocrit levels compared with WT mice, whereas mean corpuscular volume (MCV), MCH, and mean corpuscular hemoglobin concentration (MCHC) were similar (Fig. 6a). Serum concentrations of erythropoietin and bilirubin were unaltered in 24-week-old WT and *neudesin* KO mice (Fig. 6c). The increase in spleen weight and the enhanced expression of FcγR4 on RPMs were preserved in 24-week-old neudesin KO mice (Fig. 6d, g). However, the expression of FcγR1 increased compared with that at 8 weeks of age irrespective of the genotype, with little difference observed between 24-week-old WT and KO mice (Fig. 6g). These data suggest that abnormal iron recycling in youth develops into severe anemia with aging in *neudesin* KO mice.

**Fig. 5 Neudesin suppresses erythrophagocytosis and expression of FcγRs through ERK activation in bone marrow-derived macrophages. a** Scheme of experiments to measure phagocytosis of bone marrow-derived macrophages (BMDMs) in vitro. At 1 h after adding fluorescent baits, cells were collected and analyzed by flow cytometry to detect phagocytosis by BMDMs ($n = 9$). **b** Percentages of induced F4/80$^+$ macrophages from WT and *neudesin* KO bone marrow cells ($n = 9$, $P = 0.5997$ by unpaired t-test). **c** Percentage of phagocytosis by BMDMs against RBCs, apoptotic thymocytes, and killed *E. coli*. *$P < 0.05$ by unpaired t-test ($n \geq 8$, RBCs: $P = 0.0164$, apoptotic thymocytes: $P = 0.2423$, and *E. coli*: $P = 0.3560$ by unpaired t-test). **d** Pre-treatment with recombinant neudesin (100 ng/mL) for 2 days into the culture medium inhibited enhanced erythrophagocytosis by *neudesin* KO BMDMs ($n = 6$ each group, Control:WT vs. Control:KO: $P = 0.0231$, Control:WT vs. + neudesin:WT: $P = 0.9998$, Control:WT vs. + neudesin:KO: $P = 0.9994$, Control:KO vs. + neudesin:WT: $P = 0.0192$, Control:KO vs. + neudesin:KO: $P = 0.0297$, and + neudesin:WT vs. + neudesin:KO: $P = 0.997$ by a 2-way analysis of variance followed by Sidak's post-test). **e** Mean fluorescence intensity (MFI) of surface macrophage receptors on BMDMs induced from WT and *neudesin* KO bone marrow cells were measured by flow cytometry. The effects of pre-treatment with recombinant neudesin (100 ng/mL) were also investigated ($n = 6$ each group, Fcgr1; Control:WT vs. Control:KO: $P = 0.0001$, Control:WT vs. + neudesin:WT: $P = 0.371$, Control:WT vs. + neudesin:KO: $P = 0.972$, Control:KO vs. + neudesin:WT: $P < 0.0001$, Control:KO vs. + neudesin:KO: $P = 0.0003$, and + neudesin:WT vs. + neudesin:KO: $P = 0.19$, Fcgr2/3; Control:WT vs. Control:KO: $P = 0.0234$, Control:WT vs. + neudesin:WT: $P < 0.0001$, Control:WT vs. + neudesin:KO: $P = 0.0014$, Control:KO vs. + neudesin:WT: $P < 0.0001$, Control:KO vs. + neudesin:KO: $P < 0.0001$, and + neudesin:WT vs. + neudesin:KO: $P = 0.1137$, and Fcgr4; Control:WT vs. Control:KO: $P = 0.9866$, Control:WT vs. + neudesin:WT: $P = 0.3411$, Control:WT vs. + neudesin:KO: $P = 0.9993$, Control:KO vs. + neudesin:WT: $P = 0.2027$, Control:KO vs. + neudesin:KO: $P = 0.9966$, and + neudesin:WT vs. + neudesin:KO: $P = 0.2835$ by a 2-way analysis of variance followed by Sidak's post-test). **f** Cell lysates from WT and *neudesin* KO BMDMs were extracted 30 min after incubation with recombinant neudesin (100 ng/mL). Phosphorylated ERK1/2, ERK1/2, phosphorylated AKT, AKT, phosphorylated CREB, CREB, and β-Actin were detected by western blotting. **g** Relative expression of p-ERK/ERK, p-AKT/AKT, and p-CREB/CREB was determined ($n = 4$, p-ERK/ERK Control: $P = 0.0435$, p-ERK/ERK +neudesin: $P = 0.3870$, p-AKT/AKT Control: $P = 0.8851$, p-AKT/ AKT +neudesin: $P = 0.3878$, p-CREB/CREB Control: $P = 0.6681$, p-CREB/CREB +neudesin: $P = 0.8863$ by a 2-way analysis of variance followed by Sidak's post-test)). **h** Pre-treatment with MEK inhibitor U0126 (100 nM) for 2 days into the culture medium enhanced erythrophagocytosis by WT, but not *neudesin* KO BMDMs ($n = 6$ each group, Control:WT vs. Control:KO: $P = 0.0457$, Control:WT vs. + U0216:WT: $P = 0.0238$, Control:WT vs. + U0216:KO: $P = 0.0336$, Control:KO vs. + U0216:WT: $P = 0.9894$, Control:KO vs. + U0216:KO: $P = 0.9988$, + U0216:WT vs. + U0216:KO: $P = 0.9984$ by a 2-way analysis of variance followed by Sidak's post-test). **i** Pre-treatment with MEK inhibitor U0126 (100 nM) for 2 days into the culture medium enhanced surface expression of FcγR1 on WT, but not *neudesin* KO BMDMs. ($n = 6$ each group, Control:WT vs. Control:KO: $P = 0.0019$, Control:WT vs. + U0216:WT: $P = 0.0034$, Control:WT vs. + U0216:KO: $P = 0.0004$, Control:KO vs. + U0216:WT: $P > 0.9999$, Control:KO vs. + U0216:KO: $P = 0.9897$, + U0216:WT vs. + U0216:KO: $P = 0.9446$ by a 2-way analysis of variance followed by Sidak's post-test)*$P < 0.05$, and ***$P < 0.001$ by a 2-way analysis of variance followed by Sidak's post-test. Each symbol represents an individual mouse. Data are shown as means ± SD of three or more experiments.

## Discussion

Neudesin is a secretory heme-binding protein that was initially identified as a neurotrophic factor and was thereafter considered as a regulator of obesity development and energy expenditure[11]. In this study, we additionally find that neudesin plays a role in erythrocyte homeostasis by regulating splenic erythrophagocytosis.

*Neudesin* KO mice (8 weeks old) displayed mild splenomegaly with increased accumulation of RBCs without obvious anemia or polycythemia, and their peripheral RBCs were relatively tolerant to osmotic stress and sugar depletion. These data implied that senescent RBCs were removed at an earlier aging stage owing to accelerated turnover of RBCs in the spleens of *neudesin* KO mice. Concordantly, in vivo biotinylation assays demonstrated the shortened lifespan of RBCs, and cross-transfusion data excluded the possibility of faster RBC aging in KO mice. Splenectomy experiments revealed that the shortened lifespan of RBCs was caused by abnormalities in the spleen of *neudesin* KO mice. Unexpectedly, the lifespan of RBCs in splenectomized WT mice was similar to that of sham WT mice, even though the spleen is mainly responsible for removing aged RBCs. When stressed RBCs are transfused, macrophages transiently accumulate in the liver and remove stressed red blood cells as needed[26]. A similar compensatory phenomenon may be occurring in the livers of splenectomized mice.

Nevertheless, despite increased RBC clearance, we could not detect the compensatory enhancement of erythropoiesis in the bone marrow or spleen of *neudesin* KO mice. The ratios of TER119$^+$CD71$^+$ erythroblasts in the bone marrow and spleen and the concentrations of serum erythropoietin were unaltered in WT and *neudesin* KO mice (Supplementary Fig. 1). Extramedullary hematopoiesis might occur independently of erythropoietin in non-splenic tissues in *neudesin* KO mice.

It is considered that the phagocytic ability of RPMs was increased mainly by upregulated surface expression of FcγRs in

*neudesin* KO mice. CD163 molecules, which are endocytic receptors for hemoglobin-haptoglobin complexes, were also upregulated in *neudesin* KO RPMs. Previously, CD163 has been shown to mediate cell-cell interactions between macrophages and developing erythroblasts in erythroblastic islands and function as an innate immune sensor for bacteria[21,27]. It is possible that CD163 is also involved in the adhesion with RBCs and/or recognition of senescent RBCs by RPMs, and that upregulation of CD163 on *neudesin* KO RPM might contribute to the accumulation of RBCs and upregulation of erythrophagocytosis in the spleen.

The transcription factor Spi-C plays an essential role in heme-dependent differentiation from monocytes to RPMs[3,22]. *Spic* KO mice showed decreased erythrophagocytosis by splenic macrophages and increased splenic iron stores. In contrast, in addition to increased erythrophagocytosis (Fig. 3f) and decreased splenic iron store (Fig. 3h, i), *neudesin* KO mice showed upregulated expression of *Spic* mRNA in RPMs (Fig. 4d). It is possible that the upregulation of *Spic* in *neudesin* KO RPMs, which might enhance the properties of RPMs as iron-recycling macrophages, causes upregulation of erythrophagocytosis and abnormal iron homeostasis.

Neudesin is a secretory protein expressed in many tissues other than the spleen such as the nervous system and adipose tissues. We previously reported that neudesin functions as a negative regulator of sympathetic activity and affects heat production and fatty acid oxidation in brown adipose tissue and enhanced lipolysis in white adipose tissue[16]. Thus, there remains a possibility that the splenic phenotypes of *neudesin* KO mice are secondary effects from other tissues on the spleen and macrophages. However, as shown in Fig. 5e, BMDMs from *neudesin* KO, cultured independently of other tissues, exhibited upregulation of FcγRs, and recombinant neudesin protein counteracted the upregulation. Furthermore, to investigate whether neudesin

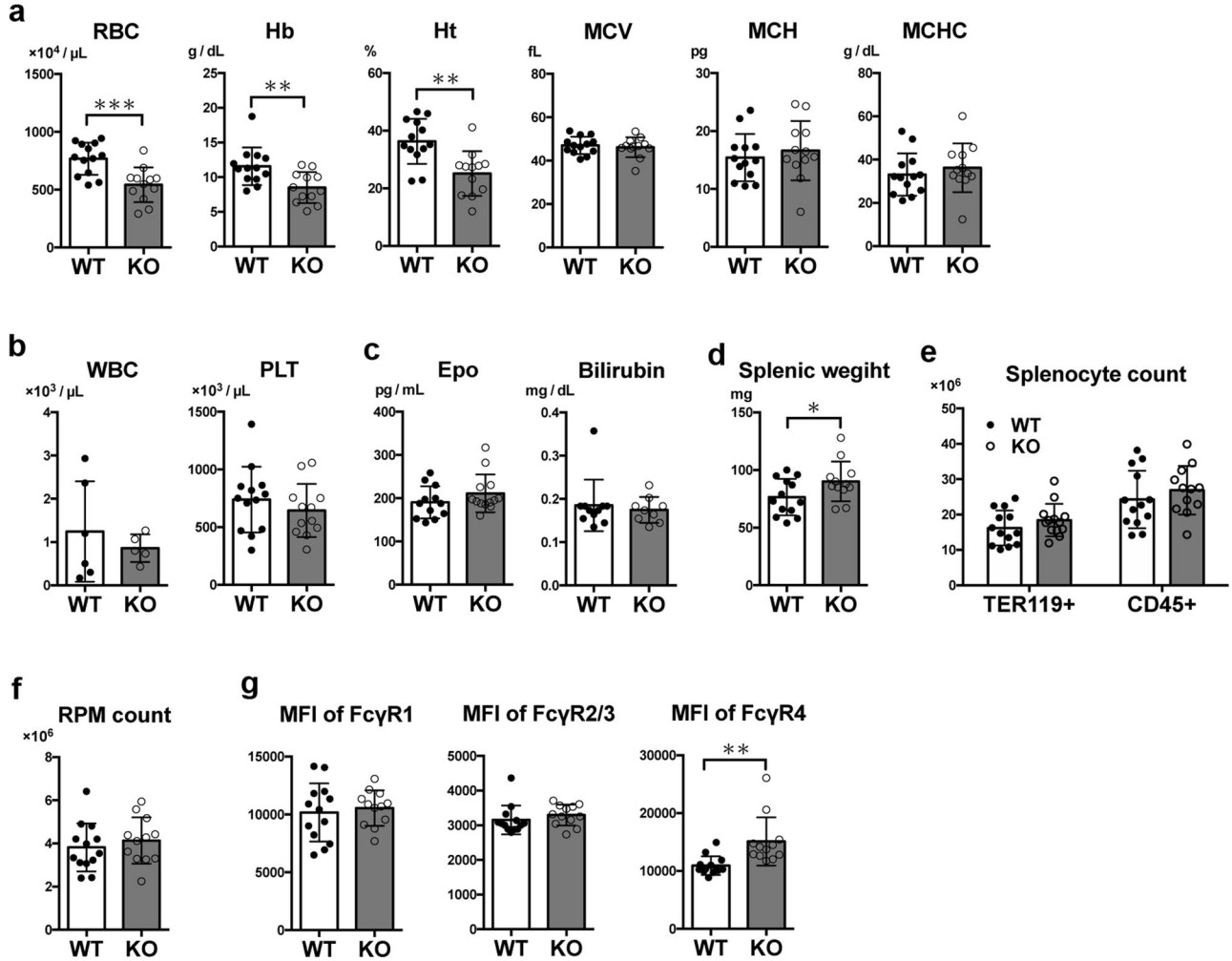

**Fig. 6 Mature neudesin-deficient mice exhibit symptoms of anemia. a, b** Blood cell counts of 24-week-old WT and neudesin KO male mice. RBC, red blood cell count (WT: $n = 16$, KO: $n = 15$, $P = 0.0193$); Hb, hemoglobin (WT: $n = 13$, KO: $n = 12$, $P = 0.0055$); Ht, hematocrit (WT: $n = 13$, KO: $n = 12$, $P = 0.0016$); MCV, mean corpuscular volume (WT: $n = 13$, KO: $n = 12$, $P = 0.6175$); MCH, mean corpuscular hemoglobin (WT: $n = 13$, KO: $n = 12$, $P = 0.5218$); MCHC, mean corpuscular hemoglobin concentration (WT: $n = 13$, KO: $n = 12$, $P = 0.4683$); WBC, white blood cell count (WT: $n = 6$, KO: $n = 5$, $P = 0.4938$); PLT, platelet count (WT: $n = 13$, KO: $n = 12$, $P = 0.3715$). **c** Serum concentration of erythropoietin (Epo, WT: $n = 12$, KO: $n = 13$, $P = 0.2198$) and total bilirubin (WT: $n = 11$, KO: $n = 9$, $P = 0.6419$) from 24-week-old WT and neudesin KO male mice. **d** Spleens of *neudesin* KO male mice were significantly heavier than those of 24-weeks-old WT male mice (WT: $n = 13$, KO: $n = 12$, $P = 0.0494$). **e** Splenocyte suspensions from 24-week-old WT and *neudesin* KO male mice were assessed by flow cytometric analysis, and cell numbers of TER119[+] red blood cells (WT: $n = 13$, KO: $n = 12$, $P = 0.2580$) and CD45[+] white blood cells (WT: $n = 13$, KO: $n = 12$, $P = 0.3981$) in the spleen were determined. **f** Chart showing the number of RPMs from 24-week-old WT and *neudesin* KO male mice (WT: $n = 13$, KO: $n = 12$, $P = 0.4752$). **g** Mean fluorescence intensity (MFI) of surface macrophage receptors on RPMs from 24-week-old WT and *neudesin* KO male mice was measured by flow cytometry (WT: $n = 13$, KO: $n = 12$, FcγR1: $P = 0.6545$, FcγR2/3: $P = 0.3510$, FcγR4: $P = 0.0027$). *$P < 0.05$, **$P < 0.05$, and ***$P < 0.001$ by unpaired t-test. Each symbol represents an individual mouse. Data are shown as means ± SD of two or more experiments.

proteins can act on the spleen in an endocrine manner from other tissues, we examined intravenous administration of recombinant neudesin proteins. A single dose or three consecutive days administration of neudesin proteins at least did not rescue the splenic phenotypes of *neudesin* KO such as increased splenic cell number and increased expressions of FcγRs on RPMs (Supplementary Figs. 2 and 3). These data suggest that, at least in the regulation of FcγRs, neudesin acts in an autocrine/paracrine manner within the spleen. Further studies using *neudesin* conditional KO mice will reveal more details about the functions of neudesin.

Our study demonstrated that neudesin, which has been identified as a heme-binding secretory factor, suppressed erythrophagocytosis of RPMs and BMDMs. The differences between WT and neudesin KO mice were small at 8 weeks of age under

normal conditions, but the small differences led to delayed recovery from drug-induced anemia and subsequent anemia in mature adulthood. The receptor for neudesin and the physiological significance of its affinity for heme remain unrevealed. Future studies should focus on further characterizing neudesin.

## Methods

**Mice**. All Animal studies were conducted in accordance with the International Guiding Principles for Biomedical Research Involving Animals and approved by the Animal Research Committee of Kobe Pharmaceutical University (approval number 2018-044, 2019-028, 2020-059, 2021-061, 2022-034, 2023-043), and the care of the animals was implemented following the Animal Research: Reporting of in Vivo Experiments guidelines. We have complied

with all relevant ethical regulations for animal use. Neudesin knockout (KO) mice were reported previously[16]. Mice were housed in a temperature-controlled pathogen-free animal facility with a 12-h light and dark cycle with free access to water and food. For in vivo study, 8- and 24-week-old male WT and KO mice in the C57BL/6N background were used.

**Quantitative RT-PCR**. RNA was extracted using the Sepasol-RNA I Super G (Nacalai tesque, Kyoto, Japan). Up to 1 µg of RNA was used for reverse transcription into cDNA and Real-time quantitative RT-PCR was performed using the THUNDERBIRD Probe qPCR/RT Set (TOYOBO, Osaka, Japan). The 18 S ribosomal RNA level was used as an internal control. The PCR primers used are listed in Supplementary Table 1.

**Flow cytometry**. Splenocytes were prepared by applying pressure to the spleen using the head of a syringe and collected through a 70-µm cell strainer. Splenocytes and blood cells were incubated with fluorochrome-conjugated antibodies and other reagents, and immunofluorescent cells were analyzed and sorted using FACSAriaIII and FlowJo software (BD Biosciences, Franklin Lakes, NJ, USA). Antibodies and reagents used are listed in Supplementary Table 2, and gating strategies are shown in Supplementary Fig. 4.

**Blood cell count**. Peripheral blood samples were harvested in EDTA-coated microtubes (Health Wave Japan, Tokyo, Japan) and analyzed with an automatic blood cell counter PCE-310 (Erma Inc., Tokyo, Japan) using the mouse species program.

**Hemolysis test**. Erythrocytes (0.4% hematocrit) were incubated at 37 °C for 16 h in Ringer solution containing 125 mM NaCl, 5 mM KCl, 1 mM $MgSO_4$, 32 mM HEPES, 5 mM glucose, and 1 mM $CaCl_2$, pH = 7.4, 305 mOsm. Hypertonic solution was obtained by adding 400 mM sucrose to the Ringer solution. In studies on glucose depletion, glucose was removed from the Ringer solution. After incubation, the samples were centrifuged for 4 min at $400 \times g$, 4 °C, and the supernatants were harvested. To quantify hemolysis, the hemoglobin concentration of the supernatant was determined photometrically at 405 nm. The absorption of the supernatant of erythrocytes lysed in distilled water was defined as 100% hemolysis.

**RBC biotinylation, transfusion, and splenectomy**. In vivo biotinylation and transfusion of RBCs followed by flow cytometry were performed at 8 weeks of age. Splenectomy was performed at 4 weeks of age, followed by in vivo biotinylation at 8 weeks of age. RBCs were labeled in vivo by intravenous injection of 30 mg/kg sulfo-NHS-biotin (Thermo Fisher Scientific). Small samples of peripheral blood were collected from the tail vein 3 days after biotin labeling and stained using Alexa Fluor 594-conjugated streptavidin. Blood samples were analyzed at 7-day intervals to quantify biotin-labeled cells in the circulation. For transfusions of biotinylated RBC, donor mice received the biotinylating reagent. Two days after injection, mice were euthanized via isoflurane overdose and promptly exsanguinated by cardiocentesis. Whole blood was collected into sterile tubes in a 1:7 mixture with anti-coagulant citrate phosphate dextrose adenine solution. After washing with PBS, 100 µL of diluted blood ($1 \times 10^9$ cells per mouse) was administered to the recipient mice via retro-orbital injection under anesthesia.

**Acute anemia model with phenylhydrazine**. Mice were intraperitoneally administered 2 mg/mouse phenylhydrazine (Sigma-Aldrich, St. Louis, MO, USA) dissolved in 200 µL PBS. At 2, 4, and 6 days after injection, peripheral blood was collected.

**Histological analysis**. Spleens were fixed in 4% paraformaldehyde in PBS for 24 h and then embedded in paraffin blocks. Paraffin sections of the spleen were prepared using a rotary microtome (RM2135, Leica, USA) and Prussian blue iron staining was performed. Iron Stain Kit (Prussian Blue Stain) (Abcam, ab150674) was used by following manufacturer's procedure. Autofluorescence of RPM was observed using an IX70 inverted fluorescent microscope (Olympus, Tokyo, Japan).

**In vivo and in vitro phagocytosis assays**. RBCs were labeled with PKH26, washed three times with PBS, and resuspended at a density of $3 \times 10^7$ cells in 100 µL in PBS. Mice were injected through the retro-orbital venous sinus with PKH26-labeled RBCs. Splenocytes were collected 5 days after injection, and stained with fluorescence-conjugated anti-F4/80 and anti-CD11b mAb. RPMs that had ingested PKH26-labeled RBCs were detected by flow cytometry. For in vitro phagocytosis assays, PKH26-labeled RBCs, PKH26-labeled apoptotic thymocytes prepared by treatment with 10 µM dexamethasone at 37°C for 4 h after PKH26-labeling, or TRITC-labeled killed *E. coli* BioParticles (Sigma) were added to BMDMs at the cell number ratio of 10:1, 10:1, and 100:1, respectively. After incubation for 1 h, cells were collected, hemolyzed, and stained with fluorescence-conjugated anti-F4/80 mAb. F4/80$^+$ BMDMs that had ingested fluorescent cells were detected by flow cytometry.

**Tissue and serum iron levels**. Serum and tissue iron concentrations were measured using a commercially available colorimetric iron quantification kit (BioAssay Systems, Hayward, CA, USA) according to the manufacturer's instructions.

**Isolation of RPMs**. RPMs were isolated from $1 \times 10^8$ splenocytes by positive selection with anti-F4/80 microbeads, MACS columns, and MACS separators according to the manufacturer's protocols (Miltenyi Biotech, Bergisch Gladbach, Germany).

**Cell culture of BMDMs**. Bone marrow cells ($2 \times 10^7$ cells/mL) were cultured in complete RPMI (RPMI-1640 medium supplemented with 10% FBS, 0.03 mg/mL L-glutamine, 100 units/mL penicillin, and 100 mg/mL streptomycin) supplemented with murine M-CSF (25 ng/mL) for 7 days, followed by stimulation with complete RPMI supplemented with 20 ng/mL murine GM-CSF and 4 µM hemin for 2 days[25].

**Western blotting**. BMDM were homogenized by sonication for 30 s in ice-cold RIPA buffer with a protease and phosphatase inhibitor (Nacalai tesque, Kyoto, Japan), and centrifuged for 10 min at $14,000 \times g$. In total, 5 µg of total protein from cell lysate was run on an 12.5% sodium dodecyl sulfate-polyacrylamide gel electrophoresis. After electrophoresis, proteins were transferred to a methanol-hydrated PVDF (polyvinylidene fluoride) membrane for 90 min and blocked with Blocking One (Nacalai tesque) for 30 min at RT. Membranes were probed overnight at 4 °C with a primary antibody. After overnight incubation, membranes were washed three times with PBST for 5 min and then incubated for 1 h at RT with an anti-rabbit IgG horseradish-peroxidase-conjugated secondary antibody. Membranes were then incubated in Chemi-Lumi One reagent (Nacalai tesque) at RT and then imaged using ImageQuant LAS 4000 (FUJIFILM, Tokyo, Japan). Protein loading was confirmed by probing for β-actin. The quantification of the western blots was performed by densitometry using ImageJ[10]. Expression levels were determined with

ImageJ software. The antibodies used are listed in Supplementary Table 2.

**Statistical analysis and reproducibility**. Data were analyzed using Prism software (GraphPad Software). Data are expressed as the mean ± SD. Unpaired *t* test and 2-way analysis of variance followed by Sidak's post-test were used as described in the figure legends. All *p* values are two-sided, and *p* values less than 0.05 were considered statistically significant. Sample sizes and number of replicates are described in the figure legends.

**Reporting summary**. Further information on research design is available in the Nature Portfolio Reporting Summary linked to this article.

## Data availability

Source data for the graphs are available as Supplementary Data file.

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

## Acknowledgements

The authors would like to thank members of the Department of Microbial Chemistry for insightful discussions and support. We thank Editage (www.editage.com) for English language editing. This work was supported by a MEXT KAKENHI Grant-in-Aid for Scientific Research (C) Grant Number 20K07060 (to Y.N.) from the Ministry of Education, Culture, Sports, Science, and Technology (MEXT).

## Author contributions

Y.N., Y.M., T. Mukae, T. Mikami, R.S., and N.K. performed the experiments, Y.N. performed data analysis, Y.N. wrote the manuscript, and H.K., N.I., and M.K. supervised the study.

## Competing interests

The authors declare no competing interests.
