## [Peer Review File · Communications Biology]

Reviewers' comments:

Reviewer #1 (Remarks to the Author):

The manuscript focuses on investigating the possible function of Neudesin in regulating splenic red pulp macrophages during erythrophagocytosis by using a Neudesin KO mouse model. The authors provided evidence based on phenotype, gene expressions and biochemical and hematological studies of the Neudesin KO mouse model. It is exciting to learn the new functional role of Neudesin in erythrocyte and iron homeostasis.

1. In this study, the mouse model is the Neudesin KO, and Neudesin is a secretory protein expressed in many other body parts. Thus, there may have a secondary effect of Neudesin KO from other parts of the body on the spleen and macrophages. I think the author should address this, at least in the discussion.

2. Is there any anemic condition in the Neudesin KO mice?

3. What are the "some environments" in Neudesin KO mouse affecting the lifespan of erythrocytes? I think the authors should discuss more about this. What is the mechanism of splenectomy leading to the recovery of the lifespan of Neudesin KO mouse?

4. Would there be a way to deliver Neudesin to the spleen of Neudesin KO mouse to investigate whether there has a compensatory effect of Neudesin on the KO macrophages?

A related question is whether there is way to express Neudesin in the in vitro cultured WT and neudesin KO bone marrow cells, so as to examine the direct effect and function of Neudesin on the differentiation and activity of BMDM cells.

5. The authors suggested ERK1/2 may be the intracellular signaling pathway targeted by Neudesin. The authors should demonstrate more on what is altered in this pathway in the absence or presence of Neudesin, and the respective effect.

Reviewer #2 (Remarks to the Author):

Neudesin is a secretory protein highly expression in the brain. Here, Nakayama et al., provide evidence for a potential novel function of neudesin in regulating splenic red pulp macrophages in erythrophagocytosis and iron recycling. This is an interesting finding. However, it may be strengthened by addressing the following questions/concerns.

1. The over-all phenotypes in neudesin KO mice appeared to be weak or subtle. Since the phenotypes were largely characterized in young adult (8-week old) mice, it would be of interest to examine if the phenotypes become sever in aged mutant mice?

2. Since neudesin, a secretory protein, is highly expressed in the brain, weak or little expressed in the spleen or macrophages, is it possible that the phenotypes in the mutant mice are largely due to the reduced circulating level of neudesin released from the brain? Is neudesin detectable in the blood? Can injecting neudesin into the mutant mice rescue the phenotypes? Can authors generate brain specific neudesin KO mice to examine the phenotypes in the spleen?

3. The data in Fig. 5f needs to be quantified. The evidence for neudesin to activate ERK pathway is not convincing.

4. Does the secretory neudesin protein bind to the spleen cells or macrophages?

Reviewer #3 (Remarks to the Author):

The manuscript convincingly shows that neudesin is a negative regulator of erythrophagocytosis by spleen red pulp macrophages. In its absence, erythroid cells are phagocytosed earlier than normal. The experiments are rigorously carried out and the data are convincing. However, while the observations on their own are interesting, the proposed mechanism is less convincing due to data interpretation as written in the manuscript. The manuscript rewritten as a technical report would be interesting to the field, however the mechanistic aspect of the paper is not as convincing.

Major concerns:

1. Interpretations of effect sizes. In several instances (Figures 1b, MCH in Table 1, splenic iron in figure 3i, figure 5e), the effect sizes are so small as to be possibly within the range of experimental error, even though there is statistical significance, likely due to high n. The authors should explain why they think these differences are biologically (not just statistically) significant, and what effect sizes would be considered as biologically significant. Genetic differences that lead to population level variances are important to report, but the authors' interpretation of the data seems to be more cut and dry. More nuance in the text with more explanation for "significance" would be important for the mechanism to be convincing. This is especially marked eg. MCH differences in table 1, where the neudesin ko is still within the normal range.
2. Related to figure 1e—is the higher deformability of the neudesin ko mice really indicative of their age, or just differences in membrane composition? Please cite or provide data, or rewrite the sentence.
3. BMDMs are used as a model for other types of macrophages in the body, however macrophages are extremely heterogenous and play many different roles in the body. Please explain how BMDMs are a good model for splenic macrophages in erythrophagocytosis (even if this is well known, an explanation would be helpful for a general audience).

KOBE PHARMACEUTICAL UNIVERSITY

4-19-1, MOTOYAMAKITA-MACHI, HIGASHINADA-KU,
KOBE 658-8558, JAPAN

To reviewer 1,

We are grateful to reviewer 1 for the useful comments and suggestions that have helped us improve our manuscript. We have responded to all the comments and suggestions. We hope that we have made the requested corrections.

Comment#1

In this study, the mouse model is the Neudesin KO, and Neudesin is a secretory protein expressed in many other body parts. Thus, there may have a secondary effect of Neudesin KO from other parts of the body on the spleen and macrophages. I think the author should address this, at least in the discussion.

Response

We thank you for pointing this out. As you pointed out, there remain several possibilities of the manner in which neudesin acts. To investigate whether neudesin proteins act on red pulp macrophages in an endocrine manner, we additionally examined intravenous administration of recombinant neudesin proteins. A single dose or three consecutive days administration of neudesin proteins did not rescue the splenic phenotypes of neudesin KO such as increased splenic cell number and increased expression of FcγRs on RPMs (Supplementary Figs. S2 and S3). On the other hand, as shown in Fig. 5e, BMDMs from neudesin KO, independent of other tissues, exhibited upregulation of FcγRs, and recombinant neudesin protein rescued the alteration. These data suggest, at least in the regulation of FcγRs, that neudesin acts in an autocrine/paracrine manner within the spleen rather than in an endocrine manner from other tissues to the spleen or a secondary effect from other parts. We added these interpretations to the Discussion section (p. 15, lines 1-18).

KOBE PHARMACEUTICAL UNIVERSITY

4-19-1, MOTOYAMAKITA-MACHI, HIGASHINADA-KU,
KOBE 658-8558, JAPAN

Comment#2

Is there any anemic condition in the Neudesin KO mice?

Response

We thank you for this question of great foresight. As shown in Table 1, 8-week-old young neudesin KO mice show no sign of anemia. In contrast, under acute anemic conditions after administration of phenylhydrazine (Fig. 2d-2f), neudesin KO mice showed decreased recovery in hemoglobin levels. Moreover, we additionally examined RBCs of 24-week-old mature WT and neudesin KO mice, and found that mature KO mice exhibited severe symptoms of anemia (Fig. 6a). We added this information to the Results (p. 12) and Discussion (p. 15, lines 21-23) sections.

Comment#3

What are the “some environments” in Neudesin KO mouse affecting the lifespan of erythrocytes? I think the authors should discuss more about this. What is the mechanism of splenectomy leading to the recovery of the lifespan of Neudesin KO mouse?

Response

We thank you for this comment and apologize for the insufficient explanation. We used the phrase "some environment" to mean that the environment surrounding RBCs other than the RBCs themselves affects the lifespan of RBCs. Subsequent splenectomy experiments showed that removal of the spleen from KO mice restored the shortened life span of RBCs, indicating that this shortened life span was caused by an abnormality in the spleen in KO mice, and that the spleen was "some environment" affecting the lifespan of RBCs. We added our interpretations to the Results (p. 8, lines 8-10 and 16-18) and Discussion (p. 13, lines 11-21)

KOBE PHARMACEUTICAL UNIVERSITY

4-19-1, MOTOYAMAKITA-MACHI, HIGASHINADA-KU,
KOBE 658-8558, JAPAN

sections.

Comment#4

Would there be a way to deliver Neudesin to the spleen of Neudesin KO mouse to investigate whether there has a compensatory effect of Neudesin on the KO macrophages?

A related question is whether there is way to express Neudesin in the in vitro cultured WT and neudesin KO bone marrow cells, so as to examine the direct effect and function of Neudesin on the differentiation and activity of BMDM cells.

Response

We thank you for the positive suggestion. As described above, we tried intravenous administration of recombinant neudesin proteins, but failed to show a compensatory effect in KO mice (Supplementary Figs. S2 and S3). Because the addition of recombinant neudesin protein in the BMDM experiments showed a compensatory effect (Fig. 5e), we suspect that the reason for the lack of compensatory effect in vivo may be the inability of the intravenously administered protein to act on the spleen.

We have tried transfection of neudesin-expressing vector into BMDMs with several transfection reagents (Lipofectamine 2000, Lipofectamine 3000, and X-tremeGENE HP). Although Lipofectamine 2000 and X-tremeGENE HP have been reported to work in BMDMs, to date we have not been able to detect expression of recombinant neudesin protein in these cells. We also tested the addition of recombinant neudesin proteins into the culture medium during differentiation of BMDMs and confirmed that the ratio of F4/80+ macrophages was unchanged.

Comment#5

KOBE PHARMACEUTICAL UNIVERSITY

4-19-1, MOTOYAMAKITA-MACHI, HIGASHINADA-KU,

KOBE 658-8558, JAPAN

The authors suggested ERK1/2 may be the intracellular signaling pathway targeted by Neudesin. The authors should demonstrate more on what is altered in this pathway in the absence or presence of Neudesin, and the respective effect.

Response

Thank you for this suggestion. It is important to explore this aspect. We examined the Jak/Stat1 pathway, which regulates CD64 expression, and found no change in the levels of Stat1 phosphorylation between WT and KO BMDMs, and treatment with the Jak inhibitor baricitinib did not alter cell surface expression of FcγRs in BMDMs. Therefore, there may be signals other than Jak/Stat1 that regulating the surface expression of FcγRs in macrophages. We have not yet identified them, and this is an issue to be addressed in the future.

To reviewer 2,

We are grateful to reviewer 2 for the useful comments and suggestions that have helped us improve our manuscript. We have responded to all the comments and suggestions. We hope that we have made the requested corrections.

Comment#1

The over-all phenotypes in neudesin KO mice appeared to be weak or subtle. Since the phenotypes were largely characterized in young adult (8-week old) mice, it would be of interest to examine if the phenotypes become sever in aged mutant mice?

KOBE PHARMACEUTICAL UNIVERSITY

4-19-1, MOTOYAMAKITA-MACHI, HIGASHINADA-KU,
KOBE 658-8558, JAPAN

Response

We thank for your suggestion. As shown in Table 1, 8-week-old young neudesin KO mice showed little to no sign of anemia. According to your suggestion, we additionally examined RBCs of 24-week-old mature WT and neudesin KO mice, and found that matured KO mice exhibited severe symptoms of anemia (Fig. 6a). We added this information to the Results (p. 12) and Discussion (p. 15, lines 21-23) sections.

Comment#2

Since neudesin, a secretary protein, is highly expressed in the brain, weak or little expressed in the spleen or macrophages, is it possible that the phenotypes in the mutant mice are largely due to the reduced circulating level of neudesin released from the brain? Is neudesin detectable in the blood? Can injecting neudesin into the mutant mice rescue the phenotypes? Can authors generate brain specific neudesin KO mice to examine the phenotypes in the spleen?

Response

We thank the reviewer for these comments. As you pointed out, some studies demonstrated that the concentration of serum neudesin was changed in human diseases. Therefore, we tried to determine serum concentrations of neudesin in WT and neudesin KO mice with commercially available and homemade anti-neudesin antibodies. However, we could not detect mouse endogenous neudesin that can be distinguished between WT and KO mouse serum. Furthermore, to investigate whether neudesin proteins can act in an endocrine manner, we examined whether intravenous administration of recombinant neudesin proteins could affect the red pulp macrophages. A single dose or three consecutive days administration of neudesin proteins did not rescue the splenic phenotypes of neudesin KO such as increased splenic cell number and increased

KOBE PHARMACEUTICAL UNIVERSITY

4-19-1, MOTOYAMAKITA-MACHI, HIGASHINADA-KU,
KOBE 658-8558, JAPAN

expression of FcγRs in RPMs (Supplementary Figs. S2 and S3). On the other hand, as shown in Fig. 5e, BMDMs from neudesin KO exhibited upregulation of FcγRs, and recombinant neudesin protein rescued the alteration. These data suggest that neudesin acts in an autocrine/paracrine manner within the spleen rather than in an endocrine manner from other tissues to the spleen. Unfortunately, owing to technical and economic reasons, it was not possible to create a neudesin cKO mouse. We added these interpretations to the Discussion section (p. 15, lines 1-18).

Comment#3

The data in Fig. 5f needs to be quantified. The evidence for neudesin to activate ERK pathway is not convincing.

Response

We agree with this. According to your suggestion, western blotting for p-ERK/ERK, p-AKT/AKT, and p-CREB/CREB were performed (Fig. 5g). The phosphorylation of ERK in KO BMDMs was weak but significantly decreased compared to that of WT BMDMs. Addition of recombinant neudesin protein tended to phosphorylate ERK, although not significantly.

Comment#4

Does the secretary neudesin protein bind to the spleen cells or macrophages?

Response

We thank you for this comment. We examined the ability of recombinant neudesin protein to bind to splenocytes by ligand binding assays with flow cytometry. Splenocytes were incubated with or without 100 ng/mL N-terminal Flag-tagged recombinant neudesin proteins at 4°C for 1 h. After incubation, these cells were stained with PE-Cy7-conjugated anti-F4/80

KOBE PHARMACEUTICAL UNIVERSITY

4-19-1, MOTOYAMAKITA-MACHI, HIGASHINADA-KU,

KOBE 658-8558, JAPAN

antibodies and Alexa488-conjugated anti-Flag-tag antibodies. As shown below, F4/80+ macrophages were slightly shifted to the right by incubation with recombinant neudesin proteins. These data together with the results from direct stimulation of BMDMs with recombinant neudesin protein, shown in Fig. 5, supports the idea that neudesin can bind to F4/80+ red pulp macrophages.

To reviewer 3,

We are grateful to reviewer 3 for the useful comments and suggestions that have helped us improve our manuscript. We have responded to all the comments and suggestions. We hope that we have made the requested corrections.

Comment#1

Interpretations of effect sizes. In several instances (Figures 1b, MCH in Table 1, splenic iron in figure 3i, figure 5e), the effect sizes are so small as to be

KOBE PHARMACEUTICAL UNIVERSITY

4-19-1, MOTOYAMAKITA-MACHI, HIGASHINADA-KU,

KOBE 658-8558, JAPAN

possibly within the range of experimental error, even though there is statistical significance, likely due to high n. The authors should explain why they think these differences are biologically (not just statistically) significant, and what effect sizes would be considered as biologically significant. Genetic differences that lead to population level variances are important to report, but the authors' interpretation of the data seems to be more cut and dry. More nuance in the text with more explanation for "significance" would be important for the mechanism to be convincing. This is especially marked eg. MCH differences in table 1, where the neudesin ko is still within the normal range.

Response

We thank you for these comments and apologize for the insufficient explanation of our interpretations. As you pointed out, the effect sizes of several experiments were small. We have changed the nuances about the points you have pointed out in the Results section (p. 6, lines 18-20, p. 9, lines 16-17, and p. 11, line 10). We added our interpretations to the Discussion section (p. 15, lines 21-23).

Comment#2

Related to figure 1e is the higher deformability of the neudesin ko mice really indicative of their age, or just differences in membrane composition? Please cite or provide data, or rewrite the sentence.

Response

We thank you for these comments. As you pointed out, alteration of the membrane composition of RBCs leads to increased or decreased deformability. On the other hand, the deformability of RBCs has been used as an indicator of age (reference 1). We added a citation and rewrote the sentence in p. 6, line 23, and p. 7, line 4.

KOBE PHARMACEUTICAL UNIVERSITY

4-19-1, MOTOYAMAKITA-MACHI, HIGASHINADA-KU,

KOBE 658-8558, JAPAN

Comment#3

BMDMs are used as a model for other types of macrophages in the body, however macrophages are extremely heterogenous and play many different roles in the body. Please explain how BMDMs are a good model for splenic macrophages in erythrophagocytosis (even if this is well known, an explanation would be helpful for a general audience).

Response

We thank you for these comments. We adopted this model because some studies listed in references 23 to 25 showed that GM-CSF and hemin-stimulated BMDMs have similar properties to those of RPMs, although they may not exactly reproduce RPMs. We rewrote the sentences on p. 10, line 21, and added the references.

REVIEWERS' COMMENTS:

Reviewer #1 (Remarks to the Author):

The authors have addressed the comments and concerns raised. The manuscript is properly revised accordingly.

Reviewer #2 (Remarks to the Author):

This revised manuscript has addressed my concerns.

Reviewer #3 (Remarks to the Author):

The manuscript is much improved from the previous version, and the authors have addressed all my concerns. In particular, the data demonstrating that the erythropoietic phenotype was not due to an endocrine function of neudesin, as well as the data with aged mice, have significantly increased the impact of the manuscript.